# Genome-Wide Identification of *CYP72A* Gene Family and Expression Patterns Related to Jasmonic Acid Treatment and Steroidal Saponin Accumulation in *Dioscorea zingiberensis*

**DOI:** 10.3390/ijms222010953

**Published:** 2021-10-11

**Authors:** Lixiu Hou, Xincheng Yuan, Song Li, Yi Li, Zihao Li, Jiaru Li

**Affiliations:** State Key Laboratory of Hybrid Rice, Department of Plant Science, College of Life Sciences, Wuhan University, Wuhan 430072, China; houlixiu1994@whu.edu.cn (L.H.); yuanxincheng@whu.edu.cn (X.Y.); songli1998@whu.edu.cn (S.L.); ly40005@whu.edu.cn (Y.L.); zihaoli@whu.edu.cn (Z.L.)

**Keywords:** *Dioscorea zingiberensis*, *CYP72A* genes, P450, jasmonic acid, phytosterol, diosgenin, steroidal saponin

## Abstract

*Dioscorea zingiberensis* is a medicinal herb containing a large amount of steroidal saponins, which are the major bioactive compounds and the primary storage form of diosgenin. The *CYP72A* gene family, belonging to cytochromes P450, exerts indispensable effects on the biosynthesis of numerous bioactive compounds. In this work, a total of 25 *CYP72A* genes were identified in *D*. *zingiberensis* and categorized into two groups according to the homology of protein sequences. The characteristics of their phylogenetic relationship, intron–exon organization, conserved motifs and *cis*-regulatory elements were performed by bioinformatics methods. The transcriptome data demonstrated that expression patterns of *DzCYP72As* varied by tissues. Moreover, qRT-PCR results displayed diverse expression profiles of *DzCYP72As* under different concentrations of jasmonic acid (JA). Likewise, eight metabolites in the biosynthesis pathway of steroidal saponins (four phytosterols, diosgenin, parvifloside, protodeltonin and dioscin) exhibited different contents under different concentrations of JA, and the content of total steroidal saponin was largest at the dose of 100 μmol/L of JA. The redundant analysis showed that 12 *DzCYP72As* had a strong correlation with specialized metabolites. Those genes were negatively correlated with stigmasterol and cholesterol but positively correlated with six other specialized metabolites. Among all *DzCYP72As* evaluated, *DzCYP72A6*, *DzCYP72A16* and *DzCYP72A17* contributed the most to the variation of specialized metabolites in the biosynthesis pathway of steroidal saponins. This study provides valuable information for further research on the biological functions related to steroidal saponin biosynthesis.

## 1. Introduction

Cytochromes P450 (CYPs) are a superfamily of heme-containing enzymes that mainly function as monooxygenases in all kingdoms of life. The genes of CYPs in the plant kingdom are usually larger than those in the kingdoms of animals and microorganisms, accounting for approximately 1% of the protein-coding genes [1]. In plants, CYPs are involved in the biosynthesis of diverse specialized metabolites, such as phytohormones, fatty acids and flavonoids, playing a crucial role in plant growth and development [2].

CYPs can be divided into many subfamilies, and CYPs belonging to the same subfamilies have the same catalytic functions in organisms. The CYPs are mainly composed of single-family clans and multiple-family clans based on phylogenetic classification [3]. The *CYP72A* gene subfamily, an important component of the multiple-family clan in CYPs, catalyzes numerous crucial reactions in the biosynthesis pathway of many specialized metabolites. Gibberellins (GAs), especially GAs with low bioactivity, exert positive influences on seed dormancy. *AtCYP72A9* has been confirmed to encode bioactive GA 13-hydroxylase in *Arabidopsis thaliana*, which contributes to the accumulation of such low-bioactivity GAs [4]. Moreover, a great number of *CYP72A* genes play indispensable roles in the biosynthesis of medicinal metabolites. In the *Medicago* genus, *MtCYP72A67* encodes a key enzyme that catalyzes hydroxylation at the C-2 position in the hemolytic sapogenin biosynthesis [5]. *GsCYP72As* have been shown to be involved in the biosynthesis of triterpenoid saponins in *Gleditsia sinensis* [6]. In addition, the expression of some *CYP72A* genes can be largely influenced by biotic or abiotic stresses. The methylation level of *LpCYP72A161* is closely connected with the response to temperature stress in ryegrass [7].

*Dioscorea zingiberensis*, a perennial vine, is mainly utilized as a source for the production of diosgenin and has attracted much more interest on account of its pharmaceutical value. Diosgenin is an indispensable precursor to steroidal hormones, and it also shows pharmacological activities against many cancers, such as liver cancer and pancreatic cancer [8,9]. However, natural diosgenin is not abundant in *D. zingiberensis**,* and it will immediately be converted into steroidal saponins by combining with aglycone structures [10]. Steroidal saponins are not only the primary storage form of diosgenin but also the main bioactive compounds in *D. zingiberensis* [11]. Steroidal saponins exhibited beneficial activities toward decreasing the risk of hyperlipidemia and cardiovascular diseases [12,13]. Previous studies have showed that the diosgenin was biosynthesized by the mevalonate acid (MVA) pathway. Acetyl coenzyme A, the initial substrate in the MVA pathway, can be subsequently converted to phytosterols and diosgenin after being catalyzed via a variety of enzymes. In fenugreek, it was proposed that the *CYP72A* gene family played a crucial role in the diosgenin biosynthesis and might participate in the biosynthesis pathway of phytosterols [14]. Moreover, enzymes encoded by the *CYP72A* gene family in legumes were reported to have a catalytic activity in the formation of triterpenoids, such as oleanolic acid, ursolic acid and betulinic acid [15,16]. Nevertheless, research on *D. zingiberensis* has mainly been focused on developing pharmacological activities [17]. Hence, it is urgent to explore the correlation between the *CYP72A* gene family and diosgenin biosynthesis.

Considering the importance of CYP72A proteins in many medicinal plants, the objective of this study was to analyze the characters of the *CYP72A* family using bioinformatics based on the genome of *D. zingiberensis*, including phylogenetic relationship, gene structure organization, conserved motif analysis, etc. In the meantime, specialized metabolites in the biosynthesis pathway of steroidal saponins (phytosterols, diosgenin and steroidal saponins) were determined, and qRT-PCR was applied to examine the expression difference of the *CYP72A* gene family under jasmonic acid (JA) treatment. Moreover, important *CYP72A* genes were screened by investigating the correlation between gene family and steroidal saponin biosynthesis. The results obtained will provide valuable insights into the function of *DzCYP72As* and identification of gene resources for breeding of *D. zingiberensis*.

## 2. Results

### 2.1. Identification of the CYP72A Proteins in D. zingiberensis

The published CYP72A protein data of *A. thaliana* were used as the reference sequence against the genomic information of *D. zingiberensis* to identify DzCYP72A proteins, and 25 sequences were identified as CYP72A proteins (Appendix A). Such identified *CYP72As* were mapped on three chromosomes (Chr1, Chr4 and Chr9) and assigned as *DzCYP72A1–DzCYP72A25* based on their genomic location (Table 1). The *DzCYP72As* were unevenly distributed across the three chromosomes, and three genes of the family were located on unknown chromosomes. The majority of DzCYP72A proteins were localized on Chr1 (*n* = 14, 56%). By contrast, Chr9 contained seven *DzCYP72A* genes (28%), and Chr4 contained only one (4%).

The detailed information and physicochemical properties of each DzCYP72A protein were predicted by the ExPASy online tool. The amino acids, molecular weight, isoelectric point, instability index, aliphatic index and grand average of hydropathicity are exhibited in Table 1. The lengths of the DzCYP72A proteins varied from 173 (DzCYP72A19) to 525 (DzCYP72A14) amino acids, with a theoretical molecular weight ranging from 19.63 (DzCYP72A19) to 60.09 (DzCYP72A25) kD. The predicted isoelectric point (pI) values of DzCYP72A proteins ranged from 6.29 (DzCYP72A19) to 9.44 (DzCYP72A14). DzCYP72A15 displayed the better thermostability while DzCYP72A7 showed relatively poor thermostability. According to the predicted results, all CYP72As were a kind of hydrophilic protein (GRAVY < 0), and most of them were unstable proteins. DzCYP72A15 was the only stable protein as its instability index was only 37.99. The subcellular localization of DzCYP72A proteins was predicted by four different online tools, and the predicted results are displayed in Table 2. Most proteins were presumed to be located in membranous organelles, such as the chloroplast, mitochondrion and endoplasmic reticulum.

### 2.2. Phylogenetic Analysis and Multiple Sequence Alignment of DzCYP72A Genes

The CYP72A proteins in *Asparagus officinalis*, *Solanum lycopersicum*, *Dioscorea rotundata* and *Oryza sativa* were identified in the same way as DzCYP72A proteins. The phylogenetic relationship among 25 DzCYP72As, 6 AoCYP72As, 13 SlCYP72As, 6 DrCYP72As, 9 OsCYP72As and 9 AtCYP72As was analyzed by MEGA7 based on the aligned results of protein sequences (Figure 1). In summary, all CYP72A proteins could be divided into two categories: monocotyledons and dicotyledons. In the meantime, the DzCYP72A family was categorized into two clades. A total of eight DzCYP72As, together with two DrCYP72As and five AoCYP72As were categorized as clade Ⅰ. Clade Ⅱ contained the remaining 17 DzCYP72As, 6 OsCYP72As and 1 AoCYP72A. Clade Ⅲ comprised four DrCYP72As and three OsCYP72As. All AtCYP72As (*n* = 9) and the majority of SlCYP72As (*n* = 12) were categorized as clade Ⅳ. Multiple alignment of DzCYP72A proteins was performed using DNAMAN8 software (Figure 2). The DzCYP72A protein sequence in clade Ⅱ had higher homology than that in clade Ⅰ. All DzCYP72A proteins could be divided into two clusters, which kept consistence with the phylogenetic results.

### 2.3. Gene Structure and Motif Analysis of DzCYP72As

The exon/intron composition of *DzCYP72As* displayed the structural diversity and complexity of this gene family (Figure 3). The exon number of *DzCYP72As* in clade Ⅰ ranged from one to six. *DzCYP72A18* had the largest gene length, and it had the maximum number of exons. In contrast, the gene length of *DzCYP72A19* was the shortest, and it had only two exons. All *DzCYP72As* in clade Ⅱ contained five exons, and *DzCYP72A5* was intron free. The coding sequence of *DzCYP72As* is listed in Appendix A.

In the meanwhile, the conserved motifs of DzCYP72A proteins were further analyzed by the MEME online search tool. A total of 10 motifs were found, and the detailed information is displayed in Figure 4. Based on the protein sequence of all DzCYP72As, we found that all DzCYP72A proteins contained motif 1, implying that motif 1 might be one of conserved motifs among all CYP72A proteins in *D. zingiberensis*. The motif number of all DzCYP72A proteins was in the range of 4–10, and motif 1 and 2 were the common composition in clade Ⅰ. The motif 1, 2, 8 and 10 were fundamental components of the DzCYP72A proteins in clade Ⅱ. The DzCYP72As in clade Ⅱ contained more motifs than those in clade Ⅰ, suggesting that such diverse motifs may be strongly correlated with corresponding functions.

### 2.4. Analysis of Cis-Regulatory Elements in the Promoters of DzCYP72As

*Cis*-regulatory elements (CREs) in the promoter region exert a crucial influence on gene functions. In order to further conduct research on the genetic functions and regulatory mechanisms of the *DzCYP72A* gene family, the 2 kb upstream sequences of 25 *DzCYP72As* were uploaded to the PlantCARE online tool. A total of 10 representative CREs (light responsive, auxin responsive, abscisic acid responsive, MYB binding site, MeJA responsive, low-temperature responsive, anaerobic induction, gibberellin responsive, salicylic acid responsive and defense and stress responsive) were visualized on each gene and are shown in Figure 5. Among all CREs, the proportion of light-responsive elements was the largest, accounting for 47%. The MeJA-responsive CREs were the second most abundant and stood at 15%. The proportions of anaerobic induction, abscisic acid responsive and MYB binding site were similar, with 9%, 8% and 7%, respectively. The proportions of auxin-responsive, gibberellin-responsive and salicylic-acid-responsive CREs were at the same level, approximately making up 3%. Additionally, the CRE related to defense and stress responsiveness occupied the least proportion, only 2%. The detailed information on CREs is exhibited in Appendix A.

### 2.5. Expression Profiles of CYP72As in D. zingiberensis

In order to further explore the functions of *DzCYP72A* genes, we analyzed the transcriptome data and investigated the tissue-specific expression patters of each *DzCYP72A* gene. The expression patterns of *DzCYP72A* genes varied by tissues (Figure 6A). *DzCYP72A23* and *DzCYP72A15* displayed the highest transcript level in both the leaf and the stem, whereas *DzCYP72A15* exhibited the highest transcript level in the rhizome. *DzCYP72A4* and *DzCYP72A5* showed the lowest expression level in the leaf, while *DzCYP72A14* represented the lowest expression level in the stem. In addition, both *DzCYP72A25* and *DzCYP72A19* exhibited the lowest expression level in the rhizome. These results implied that *DzCYP72As* may have indispensable influences on the growth and development of *D. zingiberensis*.

Numerous studies have illustrated that JA and its derivates exerted positive effects on the biosynthesis of saponins [18,19,20]. Moreover, the *DzCYP72A* gene family played a key role in the biosynthesis of steroidal saponins and the analysis of *cis*-regulatory elements suggested that *DzCYP72As* had a strong correlation with JA. Thus, the primers of *DzCYP72As* were designed (Appendix A) and qRT-PCR was applied to analyze the transcriptional expressions of *DzCYP72As* under treatment with different concentrations of JA (Figure 6B). Almost all *DzCYP72As* displayed diverse expression patterns under all treatments, but it was difficult to obtain acceptable qRT-PCR results for gene expression analysis of *DzCYP72A18*, *DzCYP72A19* and *DzCYP72A25* in rhizomes on account of their low expression abundance and amplification efficiency, which cohered with the transcriptome data.

Some genes, such as *DzCYP72A24*, *DzCYP72A13* and *DzCYP72A7*, responded rapidly and displayed higher expression levels at a relatively low concentration of JA (25 μmol/L). The transcriptional expressions of *DzCYP72A13*, *DzCYP72A14*, *DzCYP72A16*, *DzCYP72A20* and *DzCYP72A24* showed significant upregulation. The majority of *DzCYP72A* genes exhibited the highest expression level under JA with 100 μmol/L.

### 2.6. Effects of JA Concentration on the Specialized Metabolites in D. zingiberensis

Phytosterols, such as cholesterol, campesterol, stigmasterol and *β*-sitosterol, were reported as the intermediates in the biosynthesis pathway of steroidal saponins [21]. Therefore, the contents of specialized metabolites, including phytosterols, diosgenin and steroidal saponins, were determined to investigate the impacts of JA on steroidal saponin biosynthesis. It has been found that JA treatment exerted significant effects on the contents of bioactive compounds compared to those in untreated rhizomes (Table 3). Both dioscin and protodeltonin responded rapidly to low concentrations of JA, exhibiting the highest content of 38.25 μg/g and 15.69 mg/g, respectively, at JA with 50 μmol/L. The accumulation of parvifloside showed a positive relationship with the higher concentrations of JA and was approximately 1.6-fold increase over that in the control group at a dose of 100 μmol/L. Therefore, the total saponins obtained the maximum content (69.26 mg/g) at JA with 100 μmol/L, which was 1.8-fold higher than in the untreated group. Likewise, the content of diosgenin also positively responded to the JA and obtained the highest yield (647.18 µg/g) at a dose of 100 μmol/L. However, the content of cholesterol showed no significant difference under different concentrations of JA. In contrast, the yield of campesterol and *β*-sitosterol varied with different concentrations of JA, and the maximum yields of these two metabolites all occurred at a dose of 100 μmol/L. Stigmasterol obtained the maximum content at a dose of 50 μmol/L of JA, and its accumulation was decreased by relatively higher concentrations of JA.

### 2.7. Correlations among DzCYP72As, Phytosterols, Diosgenin and Steroidal Saponins

The de-trended correspondence analysis (DCA) was performed on phytosterols, diosgenin and steroidal saponins, and then was applied to calculate the maximum length of the gradient axis (LGA). Appendix A shows that the LGA value of DCA was below one. According to previous research, the redundancy analysis (RDA) is suitable for subsequent analyses when the LGA values are <3 [22]. Therefore, RDA was carried out to analyze the correlation between specialized metabolites and *DzCYP72As*. In the RDA algorithm, a total of 22 *DzCYP72As* responding to JA were used as the explanatory variables, and 8 metabolites were used as the response variables. In the meantime, a forward direction of the Akaike information criterion (AIC) was applied to further screen explanatory variables. In the first two axes of RDA analysis, these genes constrained 91.46% of variance in eight metabolites, suggesting these two axes could represent the total constrained proportion (Appendix A). As shown in Table 4, a total of 12 *DzCYP72As* had correlations with specialized metabolites of steroidal saponin biosynthesis. However, only 9 *DzCYP72As* were significantly correlated with those metabolites. Among these, *DzCYP72A6*, *DzCYP72A16* and *DzCYP72A17* displayed a highly significant correlation with such bioactive compounds. *DzCYP72A1*, *DzCYP72A3*, *DzCYP72A9*, *DzCYP72A11*, *DzCYP72A14* and *DzCYP72A20* showed significant correlations with specialized metabolites. In view of Figure 7, the screened *DzCYP72As* were positively associated with parvifloside, protodeltonin, dioscin, diosgenin, campesterol and *β*-sitosterol. In contrast, cholesterol and stigmasterol were negatively correlated with these *DzCYP72As*.

## 3. Discussion

The *CYP72A* gene family plays crucial roles in catalyzing many considerable reactions in Plantae, and many *CYP72A* genes have been cloned from various plants [4,5]. Nevertheless, genome-wide analysis of the *CYP72A* gene family has not been performed in *D. zingiberensis*, an important source of diosgenin. In this study, a total of *25 DzCYP72A* genes were identified in *D. zingiberensis* and assigned as *DzCYP72A1–25* on the base of their chromosomal location. The phylogenetic relationship, gene structure, conserved motifs, *cis*-regulatory elements and expression patterns under JA treatment were analyzed. Meanwhile, eight specialized metabolites related to the biosynthesis of steroidal saponins were determined, and the correlation between *DzCYP72As* and such metabolites was also investigated. This work provided valuable information for subsequent functional analysis of *DzCYP72As*.

The phylogenetic tree indicated that DzCYP72A proteins had stronger homology with *O. sativa*, *A. officinalis* and *D. rotundata* than with *A. thaliana* (Figure 1). *D. zingiberensis* belongs to monocotyledon and *A. thaliana* belongs to dicotyledon; therefore, *O. sativa*, *A. officinalis* and *D. rotundata*, belonging to monocotyledon, are inclined to have strong homology with *D. zingiberensis*. Moreover, the functions of CYP gene subfamilies catalyze considerable reactions in the biosynthesis pathway of numerous specialized metabolites in plants, and previous research illustrated that the *CYP72A* gene family had a profound correlation with the biosynthesis of saponins [23,24]. Hence, the reason for *D. zingiberensis* containing more *DzCYP72A* genes may be the large accumulation of steroidal saponins in the rhizome of *D. zingiberensis*. Likewise, the large amount of steroidal saponins in *A. officinalis* may be one of the reasons why all AoCYP72A proteins had strong homology with DzCYP72A proteins. Although *D. rotundata* also belongs to the *Dioscorea* genus, it contains abundant starch and only a trace amount of steroidal saponins [25]. Therefore, the large amount of starch may be closely related to the high homology of amino acid sequences between DrCYP72A and OsCYP72A proteins. In addition, *S. lycopersicum* also belongs to dicotyledon, but one SlCYP72A protein has remarkable homology with DzCYP72A proteins. According to previous research, *S. lycopersicum* contains abundant steroidal glycoalkaloids, which are also biosynthesized from phytosterols [26]. The similar biosynthesis pathway may be connected to the high homology between SlCYP72A and DzCYP72A proteins.

Similar to the case in other gene families in plants, the expression profiles of *DzCYP72As* displayed special variations in different tissues, which may be closely related to their functions in *D. zingiberensis* [27,28]. Many *DzCYP72A* genes had high expressions in the leaf and stem, but the steroidal saponins mainly accumulated in the rhizomes. According to previous research, the *CYP72A* gene family mainly participates in the biosynthesis of phytosterols, which are not only the intermediates in the biosynthesis pathway of steroidal saponins but also indispensable components for cell membranes [29,30]. Generally, the leaves have higher metabolism than rhizomes, and rhizomes cultivated for at least 3 years can satisfy the industrial requirements. Therefore, the reason for *DzCYP72A* genes having a high transcript level might be that more phytosterols are needed for maintaining the functions of cell membranes in leaves under a high metabolic rate.

Abiotic stresses, such as extreme temperature, abscisic acid, salicylic acid, JA and its derivates, exert considerable effects on plant development and the accumulation of specialized metabolisms [31,32]. In this study, the MeJA-responsive CREs were the most abundant among all CREs responsive to phytohormones, indicating that JA and its derivates profoundly correlated with the functions of *DzCYP72As*. Moreover, previous research demonstrated that JA and its derivates had positive correlations with the biosynthesis of steroidal saponins [18,19,20]. Therefore, most *DzCYP72As* were upregulated under different concentrations of JA. In most cases, MeJA was used as an elicitor in medicinal crops for enhancing the yields of bioactive compounds [33]. However, MeJA is essentially a non-bioactive compound, but it is a volatile substance, which can be helpful to increase the capacity of JA to easily enter plants via the stomata [34]. In the meantime, JA is more effective than the methylation of JA in stimulating the biosynthesis of saponins [19]. Therefore, JA was applied to the rhizomes of *D. zingiberensis* in this work.

According to previous research, Ankang of Shaanxi and Shiyan of Hubei are the original cities of *D. zingiberensis*, and the rhizomes from these two regions contained more steroidal saponins than those from other locations in China [21,35]. Therefore, seedlings from Ankang of Shaanxi were used as materials to investigate the effects of JA on *DzCYP72As* and specialized metabolites. The content of dioscin in this study was lower than that in mature rhizomes, but the contents of cholesterol, stigmasterol, *β*-sitosterol and diosgenin were higher than that in mature rhizomes [21]. Solar energy provides the source of energy for plants and is stored via photosynthesis [36]. When a plant reaches natural maturation, the aerial part of *D. zingiberensis* has already withered, inhibiting photosynthesis and energy storage. Meanwhile, the low metabolic rate and insufficient energy supply may pose negative influences on the biosynthesis of steroidal saponins, which might be one of the reasons why mature rhizomes contained only a trace amount of cholesterol, stigmasterol, *β*-sitosterol and diosgenin. Cholesterol has been proved to be one of the important compounds that provide carbon skeletons for the biosynthesis of steroidal saponins [37]. Nevertheless, the content of cholesterols had no correlation with JA in this study. In contrast, *β*-sitosterol had a profound correlation with JA. In the meanwhile, it was found that *β*-sitosterol can also biosynthesize the diosgenin and steroidal saponins [38]. Thus, the reason for the increase of steroidal saponins might be that a large amount of *β*-sitosterol participates in the biosynthesis of steroidal saponins under JA treatment. Moreover, *β*-sitosterol is also reported to be the precursor for stigmasterol, but the content of stigmasterol decreased with increased *β*-sitosterol in the high concentrations of JA, implying the accumulated *β*-sitosterol may be used for the biosynthesis of steroidal saponins [39]. In most cases, various bioactive compounds had different pharmacological activities, and parvifloside, protodeltonin and dioscin showed diverse responses to different concentrations of JA, which provided a targeted scientific basis for enhancing those metabolites. In addition, other gene families may also have strong correlations with the biosynthesis of steroidal saponins. Meanwhile, other environmental variables, such as soil salinization and moisture, can also affect expression patterns of *DzCYP72A* genes and other gene families. Therefore, further research should pay more attention to the correlation between specialized metabolites and other gene families under different environmental conditions.

## 4. Materials and Methods

### 4.1. Plant Materials

The seeds of *D. zingiberensis* were obtained from a plantation of Shaanxi Ankang in October of 2020. *D. zingiberensis* seeds were cultivated in an environmentally controlled greenhouse with 26 ± 2 ℃ and 16 h light/day. According to the manufacturer’s instruction and previous research, jasmonic acid (JA, Macklin, 98%) was dissolved in 80% ethanol and diluted in 1/2 Hoagland solution to obtain 25, 50, 100 and 200 μmol/L solutions [40,41]. Ethanol was diluted in 1/2 Hoagland solution and used as a control solution. Three-month-old seedlings with similar growth status were watered with enough control solution and JA solutions for six days. The seedlings were watered every three days, and all solutions were freshly prepared just before use. The rhizomes were collected, and every treatment was replicated three times. We finally obtained the JA-treated samples: 0 μmol/L (S1), 25 μmol/L (S2), 50 μmol/L (S3), 100 μmol/L (S4) and 200 μmol/L (S5). Some samples were promptly frozen in liquid nitrogen pending RNA extraction. While the others were freeze dried to constant weight at −80 ℃ and ground to powders with a tissue lyser.

### 4.2. Standards and Chemical Reagents

Methanol, acetonitrile, chloroform, ethanol and n-hexane were purchased from Thermo Fisher Scientific Inc. (Waltham, MA, USA), and all chemical reagents were of HPLC grade. Dioscin (98%), diosgenin (98%), *β*-sitosterol (98%), stigmasterol (95%), campesterol (98%), cholesterol (98%) and N-methyl-N-(trimethylsilyl) trifluoroacetamide were purchased from Shanghai Yuanye Bio-Technology Co., Ltd. (Shanghai, China). Parvifloside (95%) and protodeltonin (95%) were obtained from Kunming Institute of Botany, CAS (Kunming, China). Detailed information and the structures of eight compounds can be obtained on the NCBI website (https://pubchem.ncbi.nlm.nih.gov/ accessed on 30 July 2021).

### 4.3. RNA Extraction and Gene Expression Analysis

The total RNA of rhizomes was extracted using RNAiso plus (Takara, Beijing, China) according to its instructions. RNA quality and concentration were determined with the NanoDrop 2000 (Thermo Scientific, Waltham, MA, USA). Subsequently, 1 μg of total RNA was subjected to a reverse transcription reaction according to HiScript III 1st Strand cDNA Synthesis Kit (Vazyme, Nanjing, China).

All primers were designed by Primer 5.0 and are listed in Appendix A. Quantitative real-time PCR (qRT-PCR) was conducted according to the instructions of 2 × TransStart Green qPCR SuperMix (TransGen, Beijing, China). The expression patterns of the *DzCYP72A* gene family under JA treatment were analyzed using the CFX96 Real-Time PCR Detection System (Bio-Rad, Los Angeles, CA, USA). *DzActin* and *DzGAPDH* were used as internal controls. The gene’s relative expression was normalized by the 2^−ΔΔCt^ method.

### 4.4. Identification and Screening of CYP72A Family Genes

The published nine CYP72A protein sequences of *A. thaliana* were downloaded from the database (http://www.p450.kvl.dk/At_cyps/family.shtml#72A accessed on 10 July 2021) [42]. The basic local alignment search tool (BLAST) was used to identify and screen DzCYP72A proteins. In brief, these nine published AtCYP72A protein sequences were used as a query to further screen the CYP72A proteins in *D. zingiberensis,* and 25 sequences fulfilling requirements were finally identified. The genome information of *D. zingiberensis* was uploaded to the NCBI database under project PRJNA716093 (unpublished).

The genome information of *A. officinalis*, *S. lycopersicum* (TAG 3.2) and *O. sativa* (V7.0) was downloaded from the Joint Genome Institute (https://phytozome.jgi.doe.gov accessed on 10 July 2021). In the meantime, the genome information of *D*. *rotundata* was downloaded from the NCBI (project number: PRJNA695139). The CYP72A protein sequences of these species were confirmed in the same way as for DzCYP72A proteins.

### 4.5. Bioinformatics Analysis

The physicochemical characteristics of DzCYP72A proteins were predicted by an online tool, ExPASy (https://web.expasy.org/protparam/ accessed on 15 July 2021). The subcellular localization information of DzCYP72A proteins was obtained from the Plant-PLoc website (http://www.csbio.sjtu.edu.cn/bioinf/plant/ accessed on 18 July 2021), the WoLF PSORT website (https://wolfpsort.hgc.jp/ accessed on 18 July 2021), the CELLO website (http://cello.life.nctu.edu.tw**/** accessed on 18 July 2021) and the YLoc website (https://abi-services.informatik.uni-tuebingen.de/yloc/webloc.cgi accessed on 18 July 2021) [43]. The CYP72A protein sequences of *A. thaliana*, *D. zingiberensis*, *A. officinalis*, *S. lycopersicum*, *O. sativa* and *D*. *rotundata* were aligned by MEGA 7.0 and visualized using the online tool iTOL (https://itol.embl.de/ accessed on 23 July 2021). The gene structure of *DzCYP72A* was calculated by TBtools and visualized by the online website Gene Structure Display Server 2.0 (http://gsds.gao-lab.org/ accessed on 25 July 2021). [44]. The conserved motifs of DzCYP72A proteins were analyzed using the online tool Multiple EM for Motif Elicitation (MEME, https://meme-suite.org/meme/tools/meme accessed on 28 July 2021) and visualized using the TBtools [45,46]. The 2 kb upstream sequences of each *DzCYP72A* gene were predicted using the PlantCARE software (http://bioinformatics.psb.ugent.be/webtools/plantcare/html/ accessed on 30 July 2021). and presented using TBtools [47]. The transcriptome data of *D. zingiberensis* (leaf, stem and rhizome) were uploaded to NCBI database project SRR15660571, SRR15660569 and SRR15660570, respectively (unpublished).

### 4.6. Analysis of Specialized Metabolites

About 50 mg freeze-dried rhizome powder was saponified with 2 mL 2 mol/L KOH-ethanol in the water bath for one hour at 80 ℃. The unsaponifiable part was extracted using n-hexane and filtered through a membrane solution filter (0.22 μm). The supernatant was evaporated under vacuum at room temperature. Dried samples were derivatized by 50 μL N-methyl-N-(trimethylsilyl) trifluoroacetamide in room temperature for 30 min. Subsequently, 150 μL n-hexane was added pending the determination of phytosterols.

A 2:1 chloroform/methanol mixture solvent was used to extract diosgenin. Approximately 50 mg freeze-dried rhizome powder was dissolved in 1 mL of the mixture solvent and extracted by ultrasonication for 30 min. Diosgenin was obtained by centrifugation at 12,000× *g* for 10 min. The extraction of each sample was repeated three times. The supernatant extracted three times was merged and filtered through a membrane solution filter (0.22 μm) pending determination. The contents of phytosterols and diosgenin were detected using the Thermo Trace gas chromatography–mass spectrometry instrument equipped with the TG-5 ms column (30 m × 0.25 mm × 0.25 μm). The operation conditions were performed as described previously [21].

Subsequently, 80% ethanol was used for the extraction of steroidal saponins. Approximately 50 mg freeze-dried rhizome powder was dissolved in 1 mL extraction solution and centrifuged at 12,000× *g* for 10 min. This powder (50 mg) was extracted three times. The obtained supernatant was merged and filtered through a membrane solution filter (0.22 μm) for subsequent analysis. The steroidal saponins were determined by an ultra-performance liquid chromatography instrument equipped with the Q Exactive hybrid quadrupole mass spectrometer (Thermo Scientific, Waltham, MA, USA) and a reverse-phase C18 column (Thermo, 100 mm × 2.1 mm, 3 μm). The mobile phases, flow rate and temperature of chromatography column were operated as described previously [21]. The gradient program was: 0–2 min, isocratic 12% B; 2–8 min, linear gradient of 12–35% B; 8–20 min, isocratic 45% B; 20–20.1 min, linear gradient of 45–20% B; 20.1–22 min, isocratic 20% B. The data collection was performed by full scan mode (*m/z* 300–200) and selected ion monitoring mode with the diagnostic ion monitored at *m/z* 869.48.

### 4.7. Statistical Analyses

The qualitative information regarding phytosterols, diosgenin and steroidal saponins was collected from mass spectra, and the quantified results were calculated by the peak areas of external standards (Appendix A). The mean and standard deviation of each specialized metabolite were determined using SPSS 19.0 software. DCA was performed with the package vegan in R.4.0.2, and RDA was carried out on the online tool Gene Denovo (https://www.omicshare.com/tools/ accessed on 15 August 2021). The correlation between *DzCYP72As* and steroidal saponin biosynthesis was analyzed using Rstudio.

## 5. Conclusions

In this study, a total of 25 *CYP72A* genes were screened and isolated from the genome of *D. zingiberensis*. The physicochemical characteristics, subcellular localization, phylogenetic analysis, exon–intron organization, motifs, *cis*-regulatory elements and tissue-specific expression were investigated with diverse bioinformatics methods. The results of qRT-PCR and eight metabolites revealed that *DzCYP72As* and specialized metabolites displayed significant responses to JA treatment. Moreover, the total steroidal saponins, parvifloside, natural diosgenin campesterol and *β*-sitosterol were the most abundant at a dose of 100 μmol/L of JA, while protodeltonin, dioscin and stigmasterol were the most abundant at the concentration of 50 μmol/L of JA. The spearman analysis revealed that *DzCYP72As* have a strong correlation with eight metabolites in the biosynthesis pathway of steroidal saponins. The obtained results provide useful information in relation to the *DzCYP72* gene family and steroidal saponins, which is beneficial for further investigations into the evolution and functions of these genes.

## Figures and Tables

**Figure 1 ijms-22-10953-f001:**
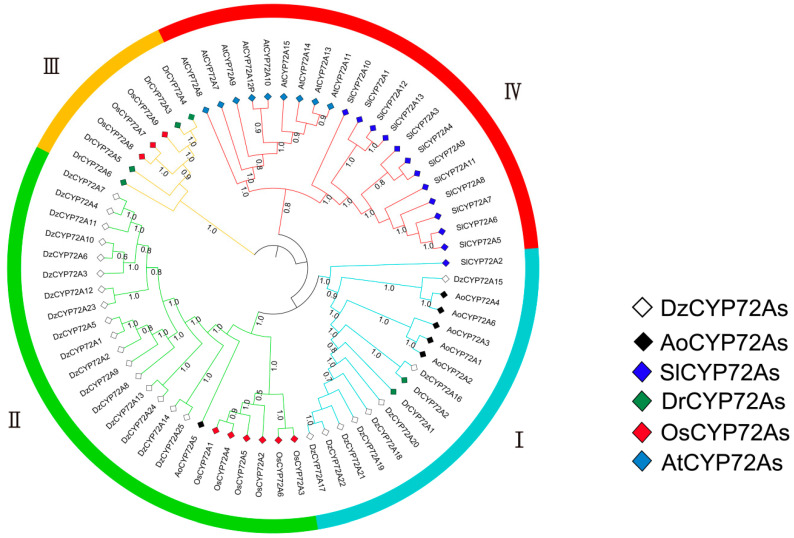
Phylogenetic tree of DzCYP72A proteins.

**Figure 2 ijms-22-10953-f002:**
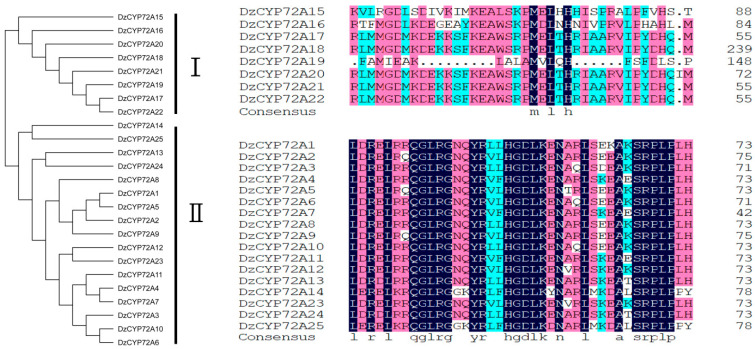
Multiple alignment of DzCYP72A proteins.

**Figure 3 ijms-22-10953-f003:**
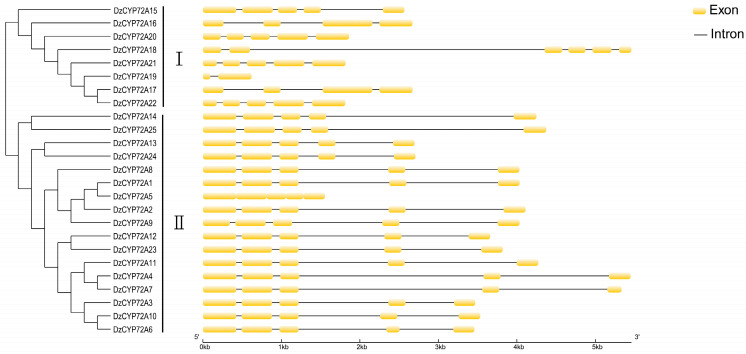
Gene structure of *DzCYP72As*.

**Figure 4 ijms-22-10953-f004:**
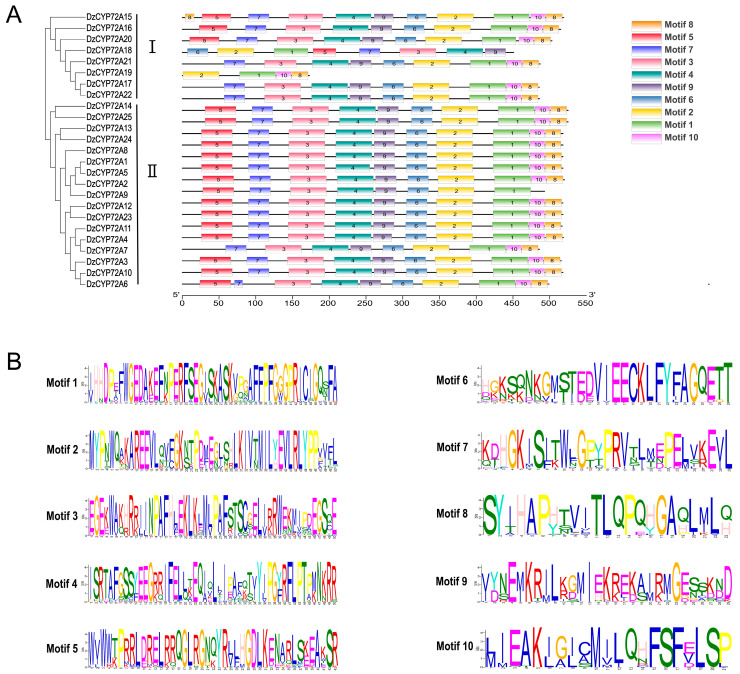
Motif pattern of DzCYP72As. (**A**) The motif composition of DzCYP72A proteins predicted by MEME and plotted in TBtools. The detailed information of ten motifs is exhibited by different colored boxes. The length of every DzCYP72A protein can be measured by the scale at the bottom. (**B**) Detailed information of ten motifs.

**Figure 5 ijms-22-10953-f005:**
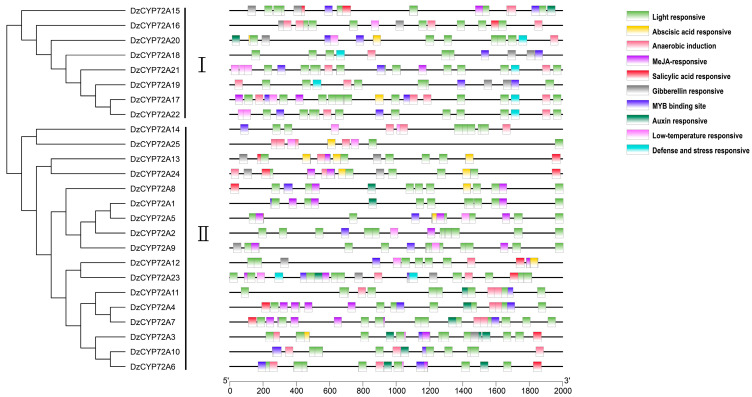
Predicted *cis*-regulatory elements in promoter regions of *DzCYPP72As*.

**Figure 6 ijms-22-10953-f006:**
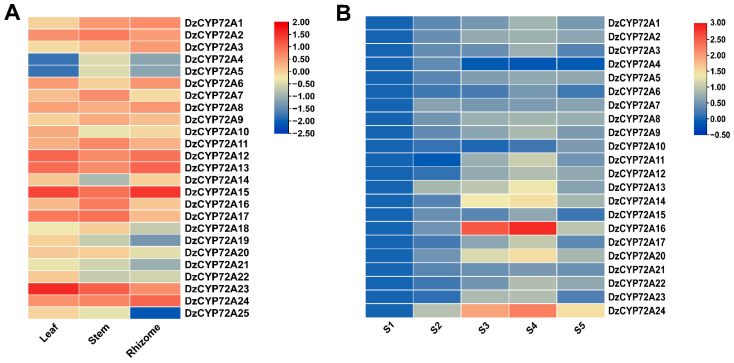
Predicted *cis*-regulatory elements in promoter regions of *DzCYPP72As*. (**A**) Tissue-specific expression profile of *DzCYP72As*. (**B**) Expression patterns of *DzCYP72As* under jasmonic acid treatment. S1: 0 μmol/L; S2: 25 μmol/L; S3: 50 μmol/L; S4: 100 μmol/L; S5: 200 μmol/L. The bar in the upper right corner represents the expression value of log10.

**Figure 7 ijms-22-10953-f007:**
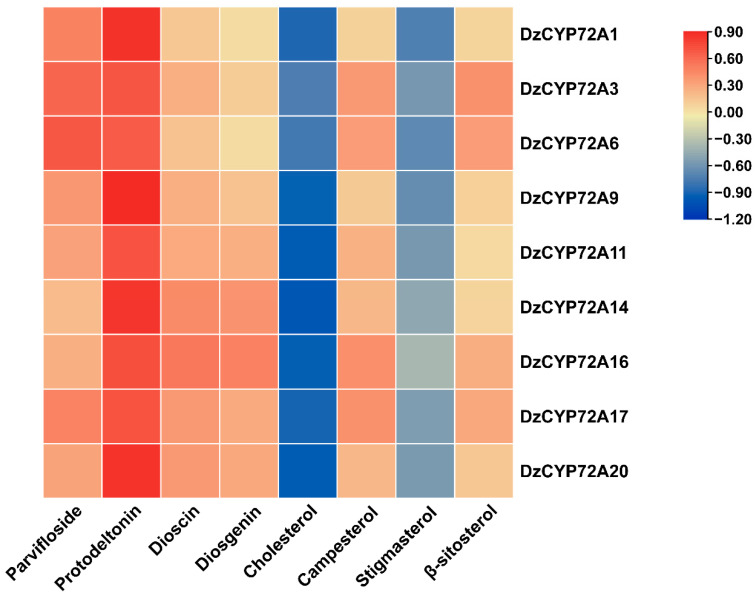
Spearman analysis of correlation between specialized metabolites and *DzCYP72As*.

**Table 1 ijms-22-10953-t001:** Characters of DzCYP72A proteins identified in *D. zingiberensis*.

Name	Amino Acids	MW (kD)	pI	Instability Index	Aliphatic Index	GRAVY	Genomic Location
DzCYP72A1	518	59.53	9.42	50.42	92.08	−0.207	Chr1:54408546–54412582
DzCYP72A2	520	59.52	9.09	46.29	93.19	−0.169	Chr1:54419523–54423635
DzCYP72A3	516	58.89	9.01	45.88	92.60	−0.192	Chr1:54482572–54486042
DzCYP72A4	519	59.74	8.98	42.74	92.02	−0.182	Chr1:54491399–54496848
DzCYP72A5	519	59.40	9.23	49.52	94.68	−0.161	Chr1:54620742–54635835
DzCYP72A6	499	56.98	8.77	48.86	91.44	−0.208	Chr1:54694341–54697800
DzCYP72A7	486	55.91	9.06	47.04	90.86	−0.241	Chr1:54703122–54708457
DzCYP72A8	518	59.46	9.16	47.69	92.82	−0.186	Chr1:54817699–54821732
DzCYP72A9	493	56.52	9.03	44.79	92.35	−0.176	Chr1:54828984–54833020
DzCYP72A10	518	59.28	9.04	46.59	93.92	−0.200	Chr1:54911925–54915459
DzCYP72A11	517	59.57	9.10	43.23	93.52	−0.162	Chr1:54920693–54924967
DzCYP72A12	518	59.21	9.27	47.44	95.98	−0.142	Chr1:54948190–54951851
DzCYP72A13	518	59.29	9.19	46.19	95.98	−0.186	Chr1:54952667–54955365
DzCYP72A14	525	60.06	9.44	46.68	93.79	−0.204	Chr1:55018259–55022509
DzCYP72A15	519	59.51	8.23	37.99	97.86	−0.093	Chr4:20462820–20465388
DzCYP72A16	515	59.04	8.36	45.35	94.45	−0.167	Chr9:1757458–1760130
DzCYP72A17	486	55.77	8.12	50.32	92.49	−0.271	Chr9:1797953–1806784
DzCYP72A18	451	51.87	8.40	46.16	91.02	−0.292	Chr9:1957935–1963395
DzCYP72A19	173	19.63	6.29	42.93	94.68	−0.070	Chr9:1963533–1964154
DzCYP72A20	503	57.97	9.13	47.24	95.37	−0.203	Chr9:2174147–2176010
DzCYP72A21	487	56.05	7.27	51.27	93.31	−0.244	Chr9:2181213–2183031
DzCYP72A22	486	55.86	8.58	50.57	91.30	−0.290	Chr9:2219226–2221041
DzCYP72A23	518	59.07	9.22	46.65	96.74	−0.121	Unknown
DzCYP72A24	518	59.36	9.11	46.56	95.23	−0.207	Unknown
DzCYP72A25	525	60.09	9.41	46.32	93.79	−0.205	Unknown

MW: molecular weight; pI: isoelectric point; GRAVY: grand average of hydropathicity; Chr: chromosome.

**Table 2 ijms-22-10953-t002:** Predicted subcellular localization of CYP72A proteins in *D. zingiberensis*.

Gene Name	Plant-PLoc	WoLF PSORT	CELLO	YLoc
CYP72A1	chloroplast	chlo: 5	mito(2.82)	chloroplast
CYP72A2	mitochondrion	chlo: 6	mito(2.38)	chloroplast
CYP72A3	chloroplast	chlo: 6	mito(2.07)	chloroplast
CYP72A4	cytoplasm	plas: 7	mito(2.06)	chloroplast
CYP72A5	chloroplast	chlo: 6	mito(2.76)	chloroplast
CYP72A6	chloroplast	chlo: 5	mito(1.7)/cyto(1.33)	chloroplast
CYP72A7	chloroplast	cyto: 9	mito(2.19)	nucleus
CYP72A8	mitochondrion	chlo: 6	mito(2.56)	chloroplast
CYP72A9	mitochondrion	chlo: 7	mito(2.08)/cyto(1.39)	chloroplast
CYP72A10	chloroplast	chlo: 8	mito(2.15)	chloroplast
CYP72A11	cytoplasm	chlo: 8	mito(2.07)	chloroplast
CYP72A12	cytoplasm	chlo: 8	mito(2.16)	chloroplast
CYP72A13	cytoplasm	chlo: 5	mito(2.59)	chloroplast
CYP72A14	cytoplasm	E.R.: 5	mito(2.97)	chloroplast
CYP72A15	chloroplast	chlo: 9	mito(1.68)/cyto(1.67)	chloroplast
CYP72A16	chloroplast	chlo: 5	cyto(1.71)	chloroplast
CYP72A17	chloroplast	cyto: 7	cyto(1.70)/mito(1.21)/nucl(1.14)	nucleus
CYP72A18	chloroplast	cyto: 9	cyto(3.09)	cytoplasm
CYP72A19	cytoplasm	cyto: 11	cyto(2.26)	cytoplasm
CYP72A20	chloroplast	cyto:11	mito(1.99)/cyto(1.42)	secreted pathway
CYP72A21	chloroplast	cyto:7	cyto(1.97)	chloroplast
CYP72A22	chloroplast	cyto: 7	cyto(1.40)/nucl(1.35)/mito(1.30)	nucleus
CYP72A23	chloroplast	chlo: 5	mito(2.21)	chloroplast
CYP72A24	chloroplast	chlo: 5	mito(2.63)	chloroplast
CYP72A25	cytoplasm	E.R.: 5	mito(2.83)	chloroplast

chlo: chloroplast; plas: plastid; E.R.: endoplasmic reticulum; cyto: cytoplasm; mito: mitochondrion; nucl: nucleus.

**Table 3 ijms-22-10953-t003:** Effects of jasmonic acid on steroidal saponin biosynthesis in *D. zingiberensis*.

Compound Name	0 μmol/L	25 μmol/L	50 μmol/L	100 μmol/L	200 μmol/L
Parvifloside (mg/g)	34.88 ± 0.85 c	39.26 ± 1.08 b	24.83 ± 1.25 d	54.89 ± 0.88 a	34.20 ± 0.55 c
Protodeltonin (mg/g)	3.81 ± 0.21 d	10.75 ± 1.51 bc	15.69 ± 1.68 a	12.37 ± 0.11 b	10.38 ± 0.25 c
Dioscin (µg/g)	19.35 ± 1.00 c	23.79 ± 0.84 b	38.25 ± 1.07 a	23.20 ± 1.07 b	11.44 ± 0.23 d
Diosgenin (µg/g)	404.30 ± 19.97 d	434.24 ± 6.37 d	595.27 ± 20.86 b	647.18 ± 24.21 a	478.64 ± 22.59 c
Campesterol (µg/g)	78.33 ± 10.92 a	76.12 ± 2.06 ab	80.53 ± 19.24 a	88.13 ± 21.41 a	51.1 ± 1.17 b
Stigmasterol (µg/g)	128.91 ± 3.92 b	117.33 ± 4.64 ab	253.94 ± 16.34 a	68.93 ± 6.93 c	69.68 ± 1.66 c
*β*-Sitosterol (µg/g)	278.62 ± 12.23 bc	299.21 ± 16.38 ab	281.27 ± 17.31 c	322.62 ± 14.77 a	226.12 ± 12.55 d

Note: The different lowercase letters represent significant difference at *p* < 0.05 (ANOVA with least-significant difference).

**Table 4 ijms-22-10953-t004:** Redundancy analysis of steroidal biosynthesis and *DzCYP72As* in *D. zingiberensis*.

Statistics	RDA1	RDA2	F	Pr(>F)
DzCYP72A1	0.9925	−0.1219	2.6747	0.0500 *
DzCYP72A3	0.9799	0.1996	2.6286	0.0417 *
DzCYP72A6	0.9486	0.3164	2.9872	0.0083 **
DzCYP72A9	0.9873	−0.1586	3.0657	0.0417 *
DzCYP72A11	0.9947	−0.1026	3.7190	0.0417 *
DzCYP72A13	0.9960	0.0896	2.5909	0.0667
DzCYP72A14	0.9838	−0.1794	4.1352	0.0167 *
DzCYP72A16	0.9993	−0.0386	4.9162	0.0083 **
DzCYP72A17	0.9997	0.0241	4.3933	0.0083 **
DzCYP72A20	0.9812	−0.1931	3.8978	0.0167 *
DzCYP72A21	0.9313	−0.3643	1.9759	0.1333
DzCYP72A22	0.9250	−0.3800	2.4202	0.1583

Note: * represents significant; ** represents highly significant.

## Data Availability

All study data are included in the main text and Appendix A.

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
