# Peer review of "Genome-Wide Identification of CYP72A Gene Family and Expression Patterns Related to Jasmonic Acid Treatment and Steroidal Saponin Accumulation in Dioscorea zingiberensis"

_ijms, 2021, doi:10.3390/ijms222010953_

Round 1
Reviewer 1 Report
Considering the high biomedical potential of the saponins contained in Dioscorea zingiberensis, the data presented in the article make a significant contribution to this area of ​​knowledge and will be of interest to a wide range of researchers engaged in the field of pharmacology, medicinal chemistry and the study of the metabolism of biologically active compounds in medicinal plants. The article can be published after revision and clarification of the following points:
1. What software was used to analyze protein sequences?
2. Was full genome or full exome analysis done?
3. What other plants of this species have been studied for the expression of enzymes of saponin synthesis, as well as genetic and phenotypic polymorphism of saponin synthesis? The fact is that certain polymorphic variants of genes are not always strongly correlated with a trait. Moreover, in plants, this process of saponin synthesis depends very much on the soil, region, frequency of watering, and the process of soil salinization.
4. In the introductory part, I would like to see the main structures of saponins contained in the plant under study, or, at least, links us to the relevant literary sources.
Author Response
Thanks for the reviewer’s comments, and please see the attachment.

Reviewer 2 Report
In this manuscript, Lixiu Hou and co-workers identified a total 25 CYP72A genes in Dioscorea zingiberensis and investigated the effect of Jasmonic acid on the expression profile of these genes. In my opinion, the manuscript may be interesting for the research community. However, there are several remarks which prevent my accepting and minor revision is required as follows:
- The supplier of JA is not listed.
- The purity of the standards and JA is not listed.
- How was the watering with JA solution performed, only once or repeatedly during 6 days?
- Did you examine the stability of the JA during treatment? Can't the solution be destabilized?
- If possible, increase the quality of figures 3, 4, and 5.
- Place Table 3 all on one page.
Author Response
Here are our responses to the reviewer’s comments point-to-point.
Point 1: The supplier of JA is not listed.
Response 1:
Thanks for the reviewer’s comments. We have added such information in manuscript. (line 350)
Point 2: The purity of the standards and JA is not listed.
Response 2:
Thanks for the reviewer’s comments. We have added such information in manuscript. (line 350; line 363-367)
Point 3: How was the watering with JA solution performed, only once or repeatedly during 6 days?
Response 3:
Thanks for the reviewer’s comments. The seedlings were watered every three days and all solutions were freshly prepared just before use. (line 354-355)
Point 4: Did you examine the stability of the JA during treatment? Can't the solution be destabilized?
Response 4:
Thanks for the reviewer’s comments. According to the manufacturer’s instruction, the purchased jasmonic acid is a very stable compound with a molecular weight of 210.27 and it will be unstable with strong oxidizing agents. Hence, we carefully prepared the required JA solutions according to its instruction and didn’t use any strong oxidizing agents. According to previous studies, researches on the effects of JA and its derivates are also carried out in this way.
In addition, methyl jasmonate is volatile and can easily enter plants by stomata, therefore it is mainly used to affect metabolites in the aboveground part. However, the steroidal saponins are abundant in the rhizome of D. zingiberensis and JA is more effective than methyl jasmonate on stimulating the biosynthesis of saponins, so the jasmonic acid (nonvolatile) was used in this manuscript.
Relevant papers:
Su, Y.N.; Huang, Y.Z.; Dong, X.T.; Wang, R.J.; Tang, M.Y.; Cai, J.B.; Chen, J.Y.; Zhang, X.Q.; Nie, G. Exogenous methyl jasmonate improves heat tolerance of perennial ryegrass through alteration of osmotic adjustment, antioxidant defense, and expression of jasmonic acid-responsive genes. Front. Plant Sci. 2021, 12:664519.
Paponov, M.; Antonyan, M.; Slimestad, R.; Paponov, I.A. Decoupling of plant growth and accumulation of biologically active compounds in leaves, roots, and root exudates of Hypericum perforatum L. by the combination of jasmonate and far-red lighting. Biomolecules. 2021, 11, 1283.
Li, J.X.; Wang, J.; Wu, X.L.; Liu, D.H.; Li, J.; Li, J.L.; Liu, S.J.; Gao, W.Y.; Jasmonic acid and methyl dihydrojasmonate enhance saponin biosynthesis as well as expression of functional genes in adventitious roots of Panax notoginseng F.H. Chen. Biotechnol. Appl. Biochem. 2017, 64 (2), 225–238.
Point 5: If possible, increase the quality of figures 3, 4, and 5. Place Table 3 all on one page.Response 5:
Thanks for the reviewer’s comments. We have placed Table 3 all on one page and increased the quality of these three figures to 800 dpi. The fuzzy pictures may be caused by the conversion of Word to PDF.